# What about Current Diversity of Mycolactone-Producing Mycobacteria? Implication for the Diagnosis and Treatment of Buruli Ulcer

**DOI:** 10.3390/ijms241813727

**Published:** 2023-09-06

**Authors:** Marine Combe, Emira Cherif, Romain Blaizot, Damien Breugnot, Rodolphe Elie Gozlan

**Affiliations:** 1ISEM, Université de Montpellier, CNRS, IRD, 34095 Montpellier, France; emira.cherif@yahoo.fr (E.C.); heydeaubeck@hotmail.fr (D.B.); rudy.gozlan@ird.fr (R.E.G.); 2Service de Dermatologie, Centre Hospitalier Andrée Rosemon, Cayenne 97306, French Guiana; romain.blaizot@gmail.com

**Keywords:** *Mycobacterium ulcerans*, DNA, cryptic biodiversity, genetic diversity, antibiotherapy, molecular testing

## Abstract

The identification of an emerging pathogen in humans can remain difficult by conventional methods such as enrichment culture assays that remain highly selective, require appropriate medium and cannot avoid misidentifications, or serological tests that use surrogate antigens and are often hampered by the level of detectable antibodies. Although not originally designed for this purpose, the implementation of polymerase-chain-reaction (PCR) has resulted in an increasing number of diagnostic tests for many diseases. However, the design of specific molecular assays relies on the availability and reliability of published genetic sequences for the target pathogens as well as enough knowledge on the genetic diversity of species and/or variants giving rise to the same disease symptoms. Usually designed for clinical isolates, molecular tests are often not suitable for environmental samples in which the target DNA is mixed with a mixture of environmental DNA. A key challenge of such molecular assays is thus to ensure high specificity of the target genetic markers when focusing on clinical and environmental samples in order to follow the dynamics of disease transmission and emergence in humans. Here we focus on the Buruli ulcer (BU), a human necrotizing skin disease mainly affecting tropical and subtropical areas, commonly admitted to be caused by *Mycobacterium ulcerans* worldwide although other mycolactone-producing mycobacteria and even mycobacterium species were found associated with BU or BU-like cases. By revisiting the literature, we show that many studies have used non-specific molecular markers (IS*2404*, IS*2606*, KR-B) to identify *M. ulcerans* from clinical and environmental samples and propose that all mycolactone-producing mycobacteria should be definitively considered as variants from the same group rather than different species. Importantly, we provide evidence that the diversity of mycolactone-producing mycobacteria variants as well as mycobacterium species potentially involved in BU or BU-like skin ulcerations might have been underestimated. We also suggest that the specific variants/species involved in each BU or BU-like case should be carefully identified during the diagnosis phase, either via the key to genetic identification proposed here or by broader metabarcoding approaches, in order to guide the medical community in the choice for the most appropriate antibiotic therapy.

## 1. Introduction

Emerging infectious diseases (EIDs) in humans are caused by a wide diversity of etiological agents, including viruses, bacteria, fungi or even parasites [1], which themselves exhibit complex inter- and intra-species genetic diversity. Interestingly, different species of pathogens or even variants of the same species can cause diseases characterized by similar clinical manifestations in the early phase of infection. For example, several viral diseases such as chikungunya and rift valley fever or influenza, nipah, SARS-CoV and MERS-CoV can all present flu-like symptoms [2]. Leptospirosis, a global bacterial zoonosis, is also characterized—at least in tropical areas—by non-specific clinical symptoms in the initial phase, such as dengue-like or other arbovirus-like syndromes, making clinical diagnosis sometimes difficult and thus delaying antibiotic therapy [3]. For such diseases it is thus necessary to develop highly specific identification tools in order to (1) identify the causative infectious agent(s) and thus support clinical diagnosis, (2) help to guide the choice of the most appropriate treatment, (3) eventually enable the development of vaccination strategies and (4) monitor transmission dynamics, pathogen circulation in the environment and disease emergence in humans.

However, the identification of an emerging pathogen by conventional methods is quite difficult and time-consuming. Whilst enrichment culture methods are laborious, require the use of appropriate media and represent a selective constraint for pathogen growth, diagnosis by serology relies on the use of surrogate antigens and is often hampered by the difficulty that a detectable level of antibodies does not appear in the blood until several days after the onset of the disease [4]. Then, culture-independent molecular tools offer—in theory—several advantages over conventional methods, including high sensitivity and specificity, speed, ease of standardization and automation [2,5,6,7]. The implementation of polymerase-chain-reaction (PCR) has resulted in an increasing number of diagnostic tests for many diseases. However, the design of specific molecular tests relies on the availability and reliability of published genetic sequences for the target pathogen, but also on the knowledge of the genetic diversity of species, subspecies or variants giving rise to the same symptoms and/or diseases. Another main limitation is that molecular detection tests are usually designed from pure laboratory isolates and are often not suitable for the identification of a specific pathogen from complex matrices such as environmental samples (i.e., soil, sediments, water, feces, etc.) [8]. However, environmental samples provide important information on the dynamics of the pathogen and the source of human contamination [4]. A key challenge is therefore to ensure high specificity of the target genetic markers in order to (1) avoid non-specific amplifications of closely related species or variants of the same species that could potentially be present in the sample and (2) to allow the simultaneous screening of a wide range of samples (clinical, environmental) to follow the dynamics of disease transmission and human emergence.

The Buruli ulcer (BU) is a human necrotizing skin disease that affects between 5000 and 6000 people annually across 33 countries worldwide, mainly located in tropical and subtropical areas [9]. BU represents the third most common mycobacterial disease after tuberculosis and leprosy [10] and is responsible for severe morbidity and mortality mainly in low-resources human populations living near contaminated water bodies [9]. Clinical signs of the disease include painless nodules, plaques and edema, followed by the development of skin ulcers that can lead to osteomyelitis and permanent disability if early detection and appropriate treatment are delayed or absent [11,12]. BU and its infectious etiological agent *Mycobacterium ulcerans* (previously described as *M. buruli*), also belonging to the mycolactone-producing mycobacteria (MPM) complex, were first described by MacCallum, Buckle, Tolhurst and Sissons in 1948 [13] from six cases of skin ulceration occurring in Australia on the basis of histology, pathogenicity, culture and microscopy of *M. ulcerans*. But the first molecular tests targeting *M. ulcerans* DNA were developed years later [10,14,15]. Since then, *M. ulcerans* is generally considered the only species responsible for BU worldwide and most studies have used three genetic markers to detect its DNA from human ulcerations, animal and environmental samples (IS*2404*, IS*2606*, KR-B). However, the genetic distinction between *M. ulcerans* and other MPM also sharing very similar clinical characteristics, namely *M. shinshuense*, *M. pseudoshottsii, M. liflandii*, *M. marinum* [16,17], based on these three commonly used molecular tests remains impossible. However, all MPM share the ability to cause a BU-like disease in humans and carefully identifying the etiological agent causing disease has major implications for BU diagnosis and the choice for the antibiotherapy. Importantly, it has been shown that *M. ulcerans* and other MPM are genetically related enough to form a single species complex, all representing variants rather than different species [17,18,19,20]. More recently, *M. shinshuense*, *M. marinum* and *M. liflandii* were found in BU-like cases diagnosed in Japan [21,22], Côte d’Ivoire (Africa) [23] and French Guiana (South America) [24], respectively. Interestingly, in Côte d’Ivoire Nguetta et al. [23] also found other mycobacteria, not belonging to the MPM complex, associated with BU-like cases, such as *M. chelonae* and *M. smegmatis*, and *M. gilvum* was found associated with some BU-like cases in French Guiana [25]. Taken together, these results suggest that (1) all MPM should definitely be considered variants of the same MPM complex and (2) all MPM variants or even other mycobacteria species apparently not producing mycolactone could cause either the same BU disease or slightly different skin ulcerations all diagnosed as BU to date.

Here we revisit the history of molecular identification tools development and assess in silico the specificity of the most commonly used genetic markers to target *M. ulcerans* DNA. We finally propose a step-by-step genetic identification key for MPM variants and discuss the potential implications of BU or BU-like skin lesions caused by other mycobacterial variants/species.

## 2. History of BU Diagnosis and *M. ulcerans* DNA Testing

### 2.1. Clinical and Microbial Diagnosis

BU was initially diagnosed by clinical observations of ulceration and confirmed by microscopic findings of acid-fast bacilli (AFB) after Ziehl–Neelsen staining (resistance to acid and/or ethanol decolorization, BAAR: bacillus acido-alcohol-resistant), a characteristic of bacteria belonging to the Genus *Mycobacterium*. The use of specific serological tests was not suitable due to cross-reactivity between mycobacterial antigens [26]. In addition, *M. ulcerans* can be difficult to culture, especially from environmental samples, due to the long generation time of the pathogen (>48 h) and proliferation of other mycobacteria [10,18,27]. Also, primary cultures usually become positive after several months of incubation and many clinical BU cases are reported as culture negative. Consequently, the implementation of rapid molecular identification tools has been a major step in the diagnosis of BU [10,15]. In addition, as BU infection has been associated with stagnant or slowly flowing waters worldwide [28,29,30], it was thought that these molecular detection tools would help understanding the environmental source of infection [31].

### 2.2. First Molecular Diagnosis

While the first molecular distinctions between *M. ulcerans,* other MPM variants and other mycobacteria species were performed by analyzing 16S rRNA, *hsp*65 and/or *rpo*B gene sequences, these genetic markers were limited to small sequence fragments and were therefore insufficient to establish a strong phylogenetic relationship between species [14,32,33]. Portaels et al. [10] and Ross et al. [15] were the first to develop PCR assays for the detection of *M. ulcerans*. Although potential candidate genetic targets had been published, there were some problems: (1) the genus-specific 65-kDa heat shock protein antigen had a high degree of homology between different mycobacterial species [34,35] and was therefore not a good target for specific amplifications [15]; (2) the 16S rRNA sequence showed only a single nucleotide divergence between the *M. ulcerans* variant and *M. marinum* [32], and it showed variability between different strains of the *M. ulcerans* [14] and was thus unsuitable for developing a simple, sensitive and specific PCR-based test. Subsequently, Portaels et al. [10] used a combination of nested PCR followed by oligonucleotide-specific capture plate hybridization (OSCPH) to “specifically” detect the *M. ulcerans* variant. First, nested PCR of a partial sequence of the 16S rRNA gene (the 5′ side of the noncoding RNA-like strand) allowed the amplification of species belonging to the mycobacteria genus. Then, OSCPH allowed the identification of a specific species from the amplified DNA product. Unfortunately, it appeared that this PCR-OSCPH test was not specific to the *M. ulcerans* variant since it also amplified genetically related *M. marinum* strains [10]. These results indicate that other MPM variants or genetically similar mycobacterial species, which have not yet necessarily been identified or described genetically, could also be amplified by this method.

### 2.3. Discovery and Implementation of IS2404 and IS2606 PCR Assays

Ross et al. [15] searched for a new and more specific DNA sequence in the *M. ulcerans* variant genome and therefore screened a genomic library of DNA fragments after digestion with AluI and HaeIII. Southern blot hybridization using a *M. ulcerans* genomic probe revealed a band of approximately 1.1 kb that was highly reactive in AluI-digested DNA, which appeared to be repeated at least 50 times in the *M. ulcerans* variant genome and therefore was considered as an ideal target for the PCR-based test [15]. This PCR test (primer pair MU1 and MU2, amplify between tandem copies of IS*2404*) was rapidly used for the diagnosis of BU human cases [15] as well as for the detection of the *M. ulcerans* variant in the environment [31]. These authors showed that the length of the complete repeat element was 1274 bp, flanked by 7-bp direct repeats with 12-bp terminal inverted repeats and a single large open reading frame potentially encoding a protein composed of 328 amino acids [36]. Their results suggested that this repeat element may constitute an insertion sequence (IS) element, which is commonly found in other mycobacteria and usually used as a template for PCR amplifications [36]. This repeat was named IS*2404* (amplified with primer pairs MU5 and MU6). They also discovered another insertion element of 1404 bp in length with 12-bp terminal inverted repeats, repeated between 30–40 times per genome, with a single open reading frame potentially encoding a protein of 445 amino acids that they named IS*2606* (amplified with primer pairs MU7 and MU8) [36]. IS*2404* and IS*2606* appeared to be unrelated to and distinct from other known mycobacterial IS, such as IS*1245* found in M. avium [37] and IS*1512* found in M. gordonae [38]. At that time, specificity tests of the IS*2404* and IS*2606* PCR assays showed that IS*2404* was specific to the *M. ulcerans* variant, while IS*2606* was also present in the genome of M. lentiflavum which is not phylogenetically related to *M. ulcerans* [36]. Importantly, it is necessary to keep in mind that at that time none of the other MPM variants were tested for the presence of IS*2404* and IS*2606* (mainly because they or their genome were unknown). It has been thus assumed that “the probability of another organism carrying both elements was low” [36]. Therefore, the medical and scientific community working on BU assumed that detection of IS*2404* and IS*2606* sequences was specific enough to ensure the unique presence of the *M. ulcerans* variant in clinical and environmental samples—and while wrong, this postulate still remains valid today.

### 2.4. Implementation of qPCR Assays

The IS-based PCR tests, particularly IS*2404*, have become the molecular basis for the detection of the *M. ulcerans* variant from which more recent studies have attempted to improve their specificity and sensitivity [39], notably through the implementation of the quantitative real-time PCR (qPCR) combined with the use of TaqMan probes that allow (1) confirmation of the identity of a DNA amplification product and (2) simultaneous amplification of multiple DNA targets [40,41] (see “Primers & Probes” in Appendix A). Indeed, Fyfe et al. [41] proposed the use of two multiplex real-time PCR assays targeting IS*2404* and IS*2606* in addition to the ketoreductase B domain (KR-B) coding sequence that is expected to be present in 15 copies in the mycolactone Pks genes *mls*A1, *mls*A2 and *mls*B. The advantage of using such multiplex PCR-based assays was to avoid false positive results due to non-specific amplifications compared with the use of a single genetic target.

## 3. On the Origin of Mycolactone-Producing Mycobacteria (MPM) Species Assignment

The historical discovery of MPM and the reasons for assigning different species names is key in our understanding of BU, as mycolactone was initially thought to be a characteristic of *M. ulcerans* only. To show that it is not a specificity of *M. ulcerans*, we go through the historical discovery of other mycolactone-producing mycobacteria with similar genomic and genetic characteristics. Based on previously published molecular evidence, we show that MPMs should not be considered as different species, but rather as variants. This has important implications for the epidemiology, diagnosis and treatment of the disease. Mycolactone is a macrolide cytotoxin with immunosuppressive properties found in the extracellular matrix surrounding large clusters of *M. ulcerans* bacilli [42]. Until 2004 it was assumed that the synthesis of mycolactone was restricted to *M. ulcerans* and depended on the acquisition of a large circular virulence plasmid called pMUM and harboring three polyketide synthase (Pks) genes such as *mls*A1 and *mls*A2 (main lactone) and *mls*B (fatty acid side chain) [16,43]. In addition, it has been proposed that plasmid acquisition may be the key event that enabled the recent emergence of *M. ulcerans* strains from a common *M. marinum* progenitor [44]. Unfortunately, other mycobacteria were later found to harbor such plasmid and also produce mycolactone.

In 2004, a previously unknown mycobacterium named *M. liflandii* was isolated from the West African *Xenopus tropicalis* frogs that were housed in the laboratory with *Xenopus laevis* colonies to which the mycobacterium spread [45]. This new mycobacterium appeared to harbor IS*2404*, IS*2606* as well as a version of the pMUM plasmid and produced mycolactone E [43]. Sequence comparisons of *hsp*65, the 16s rRNA gene, the internal transcribed spacer (ITS) of the 16S-23S rRNA gene and fragments of the *rpo*B gene showed that *M. liflandii* shared >98% nucleotide identity with *M. ulcerans* and *M. marinum* [45]. A year later in 2005, another mycobacterium was isolated during an epizootic of mycobacteriosis in striped bass (*Morone saxatilis*) in Chesapeake Bay, harboring IS*2404*, IS*2606*, a pMUM-like plasmid with *mls* genes, producing mycolatone F [46] and sharing >98% nucleotide sequence identity with *M. ulcerans*, *M. marinum* and *M. shottsii*. However, it was given a new species name *M. pseudoshottsii* based on slightly different phenotypic traits such as photochromogenicity and lack of growth at 37 °C [47]. Similarly, Ucko et al. [48] found a group of *M. marinum* strains isolated from diseased fish in Israel that were positive for *mls* genes and produced mycolactone F [46]. In Japan, the first case of *M. shinshuense* infection was described in 1980 [49] and although showing some phenotypic and molecular variability it was closely related to *M. ulcerans* [50], with the presence of multiple copies of IS*2404*, the pMUM plasmid and mycolactone production [51]. Subsequently, all mycolactone-producing mycobacteria were collectively grouped into the mycolactone-producing mycobacteria (MPM) complex (Table 1), thus including *M. ulcerans*, *M. shinshuense*, *M. liflandii*, *M. pseudoshottsii* and *M. marinum* (with the exception of the non-mycolactone-producing strains of *M. marinum*).

At that time, all MPM were considered different species based on phenotypic characteristics (e.g., colony morphology, growth rates) and limited genomic background (e.g., 16S rRNA, *hsp*65, *rpo*B). However, based on the definition of prokaryotic species all MPMs sharing >98% 16S rRNA identity, >70% DNA-DNA hybridization (DDH), harboring a pMUM plasmid, containing IS*2404* and producing mycolactone should have been considered as variants of the same MPM complex rather than different species [19]. As described above, all MPMs harbor a mycolactone-producing virulence plasmid and multiple copies of IS*2404* [44,52]. Portaels et al. [14] found that *M. shinshuense* shared similar nucleotide modifications to *M. ulcerans*, including a G at position 1248 of the 16S rRNA that has been initially described as characteristic of *M. ulcerans*. Yip et al. [16] conducted a DDH-based study to investigate the relationship between MPM and *M. marinum* and showed that all MPM have a relative binding ratio (RBR) of 88–100% whereas they only have an RBR of 15–60% when compared to non-mycolactone-producing *M. marinum* strains. In addition, by analyzing large sequence polymorphisms, Kazer et al. [53] showed a clear clustering of MPM strains that were assigned different species names.

All these results converge to the same point and suggested years ago that all MPM represent genetic variability (variants) from the same MPM complex rather than different species, but also that these variants have a wider geographic distribution and host range than initially assessed (infecting fish, frogs and humans worldwide) [21,22,24] as well as a broader zoonotic potential [16,19]. This was confirmed by the association of *M. shinshuense*, *M. marinum* and *M. liflandii* with human skin ulcerations diagnosed as BU in Japan [21,22], Côte d’Ivoire [23] and French Guiana [24]. Based on this evidence, we suggest that all MPM should definitely be considered as variants from the same complex. We will use this terminology later.

## 4. Lack of Specificity of (q)PCR-Based Tests for *M. ulcerans* Variant Detection

### 4.1. Lack of Specificity of IS2404, IS2606 and KR-B

Unfortunately, all these (q)PCR-based tests have failed to specifically detect the *M. ulcerans* variant and thus exclude the presence of other MPM variants in samples. Fyfe et al. [41] showed that even their TaqMan multiplex-qPCR, which was intended to be highly sensitive and specific, resulted in positive DNA amplifications for several MPM variants, including 12 strains of *M. ulcerans*, *M. liflanddii* strain 128FXT, *M. pseudoshottsii* strain L15, and *M. marinum* strain DL045, strain CC240299 and strain DL240490. To validate these results, we performed an in silico genomic analysis to look for the presence and the copy number of IS*2404*, IS*2606* and KR-B when using different (q)PCR based assays to detect the *M. ulcerans* variant. For this purpose, we collected a set of complete genomes and pMUM plasmids available for various variants and/or strains of MPMs. We included some *M. marinum* strains as controls and *M. gilvum* was also used as an outgroup since this mycobacterium was recently found to be associated with BU-like cases in French Guiana (South America) although it does not belong to the MPM complex [25]. We were therefore interested in the potential presence of ISs and the KR-B domain in its genome and/or plasmids that could explain its association and pathogenicity in humans (See “Genomes & Plasmids” in Appendix A). As previously shown, we found that none of the IS*2404* (q)PCR-based assays were specific to the *M. ulcerans* variant since IS*2404* was present in multiple copies in the genomes and plasmids of MPM variants tested here, such as *M. liflandii*, *M. pseudoshottsii*, *M. shinshuense* and *M. marinum* DL24090 (Figure 1A,B). Furthermore, we observed that the number of copies of IS*2404* per genome was variable depending on the primer pairs used (Figure 1A), while the copy number per plasmid was lower but more stable between the different molecular tests (Figure 1B). Also, the KR-B domain was found in the plasmid of *M. ulcerans* strain Agy99 (15 copies), *M. liflandii* (14 copies) and *M. shinshuense* (15 copies) (Figure 1B). However, it is well known that genetic sequences contained in bacterial plasmids, as is the case for the KR-B sequences of pMUM, can easily be transferred to other bacterial species by horizontal gene transfer (HGT) [18] and therefore do not represent good candidates for species identification. Almost all *M. marinum* strains tested in silico harbored IS*2404*, IS*2606* or KR-B, except *M. marinum* strain DL24090 that harbored IS*2404* and IS*2606* in its plasmid but lack the KR-B domain (then the distinction with other MPM variants relies only on the absence of KR amplifications). Interestingly, none of these mobile genetic elements were found in *M. gilvum*, raising the question of its mechanisms of pathogenicity in humans. These in silico results are thus in agreement with the recent detection of *M. gilvum* in BU-like biopsies by metabarcoding when IS*2404* and KR-B DNA amplifications and Lowenstein–Jensen (LJ) culture were all negative [25].

All together these results suggest that we can definitively consider that all studies that focused on IS*2404*, IS*2606* and/or KR-B DNA amplification to detect the *M. ulcerans* variant from patient or environmental samples could not distinguish with certainty between MPM variants, including *M. ulcerans*, *M. shinshuense*, *M. liflandii* and *M. pseudoshottsii*, neither for some *M. marinum* (notably strain DL24090) when studies did not targeted KR-B.

### 4.2. Lack of Reliability of IS2404 and IS2606 CT-Values

Since the molecular targets IS*2404*, IS*2606* and KR-B are also present in the genome and/or plasmid of the MPM variants, Fyfe et al. [41] proposed that then the distinction between the *M. ulcerans* variant and other MPM variants could potentially be made on the basis of the difference between the number of IS*2606* CT-values (CT for cycle threshold) and the number of IS*2404* CT-values. In fact, while the *M. ulcerans* variant was expected to have a high copy number of IS*2606* repeats, other MPM variants were found to have fewer or no IS*2606* copies per genome [41]. Our in silico genomic analysis confirms a higher copy number of IS*2606* per genome in *M. ulcerans* strains while other MPM variants showed 1 or 0 copies per genome or per plasmid (Figure 1A,B). Therefore, why have only few studies used the difference between the number of IS*2606* CT-values and the number of IS*2404* CT-values to discriminate the *M. ulcerans* variant from other MPM variants? The answer maybe lie in several potential biases: (1) the IS*2606* test is much less sensitive than the IS*2404* test due to the presence of fewer target copies per genome (reliable detection limit of 9 copies of IS*2606* vs. 2 copies of IS*2404* in the sample) [41]; (2) the lack of knowledge of the number of ISs per genome for each MPM variant/strain present in the sample, which, as demonstrated here, is variable depending on the (q)PCR-based test used (Figure 1A); (3) the need to discard PCR enzyme inhibitors (such as salts, polysaccharides, humic acids or even urea) which is random from sample to sample.

Overall, despite the obvious non-specificity of these (q)PCR-based tests, they remain the gold standard for detection of *M. ulcerans* variant from clinical and environmental samples worldwide. Importantly, most studies on BU have ignored—or at least underestimated—the possibility of other MPM variants as potential causative agents of BU disease or the existence of very similar skin ulcerations all named BU to date.

## 5. Implementation of MIRU-VNTR to Assess Genotypic Diversity

In order to improve our knowledge of the epidemiology of BU as well as to better identify the source of human infection, several authors have attempted to screen genotypic diversity among strains of the *M. ulcerans* variant of various origins: (1) 16S rRNAs PCR-restriction profiling (AFLP) analyses revealed four different profiles of *M. ulcerans* strains such as African, South-East Asian, Mexican and South American strains [55]; (2) sequencing of the 3′ end of the 16S rRNA classified the *M. ulcerans* strains into five groups according to their geographic origin (Africa, Australia, Mexico, Asia, South America) [14]; (3) IS*2404* restriction fragment length polymorphism (RFLP) fingerprinting divided the isolates into six groups corresponding to 6 geographical regions (Africa, Australia, Mexico, South America, Asia, Southeast Asia) [56]; (4) the 2426 PCR-based assay identified nine distinct PCR profiles among *M. ulcerans* isolates that also correlated with their geographic origin [27] (Stinear et al., 2000a). While these molecular tools were able to distinguish *M. ulcerans* strains from distinct geographical sources, they were not informative about intra-variant genetic variability within the same geographic area. Furthermore, they were all limited to the study of clinical samples or bacterial cultures since they differentiated *M. ulcerans* strains on the basis of multiband profiles that are not compatible with the non-specific (multi-band) amplifications usually found in the molecular screening of environmental samples (water, sediment, etc.). It is also important to note that the specificity of these methods has not been assessed by genotyping other MPM variants.

### 5.1. Implementation of MIRU-VNTR Typing

Variable number of tandem repeats (VNTR) typing is a PCR-based test that relies on the presence of short intergenic and polymorphic DNA regions (10- and 100-bp in length) repeated in variable copy number (tandem repeats, TRs) in the genome of monomorphic species [57,58,59]. TRs have been identified in *M. tuberculosis*, *M. bovis* and *M. avium* and mycobacterial interspersed repeat units (MIRUs) and other VNTRs have been shown to be informative multilocus genotypic markers to differentiate subspecies [60,61]. Stragier et al. [57] were the first to investigate whether MIRU-VNTR typing could be suitable to differentiate and subtype *M. ulcerans* variant and *M. marinum* variant/strains. Although they found seven polymorphic MIRU-VNTR loci in both genomes, their genotyping was only able to discriminate between *M. ulcerans* variant and *M. marinum* variant/strains from different geographical origins. These tests showed a lower discriminatory capability than the previously proposed PCR 2426 [27] and limited intra-variant genetic differentiation [57]. Here again, none of the MPM variants were tested, with the exception of the *M. ulcerans* variant. Furthermore, this test required the combination of 4 MIRU-VNTR locus (MIRU locus 1, MIRU locus 5, MIRU locus 9, MIRU locus 33) to identify seven distinct *M. ulcerans* profiles although MIRU1 could not be amplified for *M. ulcerans* isolates from Surinam and French Guiana, thus excluding this locus for further analysis of South American strains (see “MIRU-VNTR” in Appendix A). When we examine the presence and copy numbers of MIRU5 in the complete genomes of several MPM variants, we can observe that *M. ulcerans* and *M. shinshuense* variants share a similar pattern, while MIRU9 seems not to be useful to discriminate between MPM variants or *M. marinum* (Appendix A). At the same time, and since none of the MPM variant genomes were available, Ablordey et al. [58] searched for TRs loci in closely related *M. marinum* genomes and found a total of 12/19 polymorphic loci also in *M. ulcerans*. They were able to identify eight *M. ulcerans* variant genotypes, all of which clustered again by geographical origin. No intra-variants and/or intra-species variability could be identified. However, their results showed that the VNTR 18 and 19 loci showed the greatest variability, potentially distinguishing between *M. ulcerans*, *M. shinshuense* and *M. pseudoshottsii* variants (Appendix A). Although their results indicate that VNTR locus 6 and locus 14 could be used to discriminate between Surinamese and French Guiana isolates of *M. ulcerans*, we did not observe different profiles for VNTR6 among the tested isolates nor between French Guiana *M. ulcerans* and *M. liflandii*, *M. pseudoshottsii* or *M. shinshuense* variants (Appendix A). Interestingly, the VNTR ST1 combination with MIRU1 has been proposed to differentiate clinical isolates of the *M. ulcerans* variant from Ghana into three VNTR allelic combinations [59]. Unfortunately, this type of VNTR combination (ST1-MIRU1) cannot be applied to all geographic areas as MIRU1 is not amplified by all *M. ulcerans* variants. Here we analyzed only one variant isolate from Ghana (MU.Agy99) but we have also found the presence and the same number of ST1 repeats into the genome of *M. ulcerans* variant isolate from Côte d’Ivoire (MU.CSURQ0185) and the Democratic Republic of Congo (MU.SGL03) as well as in the genomes of *M. liflandii*, *M. pseudoshottsii* and *M. shinshuense* variants (Appendix A). To further analyze genotypic variability, Hilty et al. [62] identified an additional 34 VNTR loci in the complete *M. ulcerans* genome. This analysis allowed them to define subgroups (designated i–iv) consisting of several loci or a single locus, but they were not able to clearly discriminate between strains from the same geographic area. Furthermore, many of these VNTR loci are absent from the genomes of the MPM variants tested, while those that are present may not be sufficient to discriminate the variants (Appendix A). Finally, the VNTR SSR (minisatellite) showed heterogeneity reflecting the geographical origin of *M. ulcerans* isolates, while no other MPM variants were tested [63]. 

### 5.2. Main Limitations and Bias

Many studies have used a combination of four VNTRs such as MIRU1, VNTR6, VNTR19 and ST1 to discriminate between *M. ulcerans* and other MPM variants in Ghana, not only from patient samples but also from environmental samples such as water, soil, biofilm and aquatic invertebrates [8,64,65,66,67]. In Australia, patient isolates, aquatic plant biofilm, mosquitoes, water filtrates and also marsupial feces were genotyped using a combination of 13 VNTRs, although the authors acknowledged that some loci (notably MIRU1 and MIRU33) generated non-specific amplifications and therefore false positives in all reactions tested and most notably for environmental samples [68]. Their results indicated that VNTR amplification was possible from clinical samples when at least ≥10 genomes µL^−1^ of purified genomic DNA extracts were present in the sample. Whereas for environmental samples, DNA concentrations had to reach a minimum of ≥100 µL^−1^ genomes. Such a high concentration can be explained by the presence of enzyme inhibitors, DNA integrity and the presence of a huge amount of DNA from other organisms [68]. Therefore, few environmental samples from Victoria, Australia contained sufficient amounts of DNA to be efficiently genotyped using VNTR amplifications and the authors recommended caution in interpreting these data due to the frequency of non-specific amplifications observed. In Côte d’Ivoire, the same combination of VNTRs was used to discriminate between isolates of the *M. ulcerans* variant, *M. marinum*, *M. chelonae* and *M. smegmatis* [23]. In French Guiana a different combination of VNTRs (VNTR18, VNTR19, MIRU33, MIRU5, ST1, SSR) had to be used to genotype *M. ulcerans* variant isolates, revealing an uncommon high genetic diversity among *M. ulcerans* variant isolates of the same geographical origin [69]. Using a combination of six MIRU-VNTR loci on tissue specimens or mycobacterial cultures, Stragier et al. [54] defined specific VNTR profiles for all *M. ulcerans* variant isolates (0 repeat at MIRU locus 2; 1 repeat at MIRU locus 5) and a unique profile corresponding to the *M. liflandii* variant. However, *M. pseudoshottsii* and *M. marinum* variants from Israel and Greece (belonging to the MPM complex) shared a common profile and the *M. shinshuense* variant was not included in the analysis.

Although VNTR genotyping has some advantages, such as being inexpensive, being able to analyze multiple samples at the same time (multiplex PCR) and requiring little DNA load taken directly from a tissue sample, it has also been suggested that VNTR typing targets unstable repetitive elements of the *M. avium* subsp. *paratuberculosis* genome that may be too unpredictable to draw accurate conclusions about genetic diversity and the relationship between isolates [70]. This tool still has major limitations in the study of BU epidemiology and *M. ulcerans* variant transmission pathways. Firstly, the resolution of the method is cumulative and therefore requires combining multiple loci to define species- or variant-specific DNA profiles, leading to laborious genotype assignments. Secondly, some discrepancies in amplicon size can result from inaccuracies in the available genomes, leading to major problems in strain/variant identification [58]. Thirdly, the VNTR combinations are not homogenized between studies, making it difficult to compare the genetic diversity between variants of *M. ulcerans* from different geographical areas. Fourth, the methodology assumes that only one mycobacterial species or MPM variant is present in the sample analyzed. However, and by contrast to previous considerations, other unidentified mycobacterial species (*Mycobacteria* sp.) have been found in biopsies of BU or BU-like from French Guiana and associated with the presence of the *M. ulcerans* variant and *M. gilvum*, also suggesting potential co-infections [25]. Finally, the test may not be suitable for environmental samples in which several mycobacterial species and/or MPM variants could be present. In this case, VNTR DNA amplifications could show multi-banded DNA profiles, also due to several non-specific amplifications, which would make it difficult or impossible to assign specific genotypes to a species or MPM variant. Furthermore, the consensus sequence resulting from the sequencing of the DNA amplifications would underestimate the true genetic diversity of the mycobacteria present in these samples.

### 5.3. The Use of a Universal Identification Key

We suggest that in order to obtain comparable VNTR profiles between future studies and thus comparable and reproducible strain genotyping it would be necessary to apply exactly the same VNTR analysis to all samples. With regards to this objective and based on previous results, we propose a universal identification key of species and/or MPM variants based on a hierarchical classification of copy number variations of the MIRU-VNTR loci (Figure 2). Whilst based on an in-silico analysis, this key still has to be laboratory validated by using both clinical and more complex, and thus more challenging, environmental samples. Overall, we suggest that a combination of metabarcoding and NGS approaches should be considered in order to capture the true inter- and intra-species diversity in both clinical and environmental samples.

## 6. Implication for the Diagnosis and Treatment of BU

### 6.1. Implication for BU Diagnosis

BU has a wide spectrum of clinical manifestations [71], with extremely aggressive infections in some areas while others are indolent or rapidly healing [14]. These observations have raised the hypothesis that if, in a given area, *M. ulcerans* is genotypically homogeneous, then host factors such as immunity, metabolism and nutrition may play an important role in the history of infection [14]. However, while MPM variants were initially given different species names based on slightly different phenotypic characteristics, they all show great genotypic similarities making it clear that they represent rather variants of the same MPM complex. This means that a variety of MPM variants are capable of causing similar skin ulcerations all named BU, a scenario that could also explain the spectrum of clinical observations. This hypothesis is notably supported by the association of clinically diagnosed BU cases with *M. liflandii or M. liflandii*-like variants [24,43] or *M. marinum* variants [57]. In these cases, IS*2404* and KR-B PCR were both positive. While some studies reported negative cultures for the *M. liflandii* variant [43] others described positive cultures with phenotypic features similar to the *M. ulcerans* variant such as slow growth (>6 weeks), yellowish coloration and granular appearance of the colonies [24]. Then, whilst *M. liflandii* and *M. pseudoshottsii* variant infections have been mainly reported in frogs and fish, respectively, these germs may be under-detected in human lesions and should be looked for in order to determine their pathogenicity. Nevertheless, it is also important to keep in mind that although *M. marinum* variant infections can be mistaken for BU, they usually present specific clinical syndromes such as verrucous plaques or sporotrichoid lesions which are not typical of *M. ulcerans* variant infections [72]. It is also noteworthy that even in lesions mimicking stricto sensu BU, *M. marinum* variant lesions do not present undermined edges, which are the most specific sign of *M. ulcerans* variant infections. Therefore, it is necessary to bear in mind that some strains of *M. marinum* can also be responsible for a different nosological entity (the so-called “fish-tank disease”). Also, whilst *M. shinshuense* and *M. ulcerans* variant infections are characterized by similar clinical presentations, an aquatic exposure does not seem to be relevant in *M. shinshuense* variant transmission [73].

Has BU been underestimated? Since clinical diagnosis is usually based on careful examination of the ulceration and then confirmed either by IS*2404* PCR amplification and/or direct microscopy and/or histopathology (including AFB staining, a characteristic shared by all mycobacteria; [74]) and/or culture, it seems that the answer is no. However, the genetic diversity of the mycobacterium capable of causing BU or BU-like disease has been underestimated.

### 6.2. Implication for M. ulcerans Genetic Diversity

Another question remains: do these MPM variants represent the genetic diversity of *M. ulcerans*? Some authors have suggested that the *M. ulcerans* variant has recently passed through an evolutionary bottleneck and adapted to a new niche environment [18]. Tobias et al. [17] referred to “niche-adapted” mycobacteria, with *M. liflandii* and *M. ulcerans* variants representing thus different ecotypes. Interestingly, we found both variants in the same environmental habitats (sediments) in French Guiana as well as in BU biopsies [24], suggesting that these MPM variants share the same environments. Since BU is known to be caused by the *M. ulcerans* variant worldwide, we propose to consider that all other variants belonging to the MPM complex represent *M. ulcerans’* variability. Nevertheless, we do not exclude that within this MPM complex the *M. ulcerans* variant did not appear first but rather evolved from another ancestor variant. Notably, based on the number of ISs repeats (IS*2404* and IS*2606*) and the number of pseudogenes present in the MPM genomes, we suggest that they could have evolved from an ancestor variant to adapt to the host they infect rather than to an ecological niche (Figure 1). For instance, the *M. ulcerans* variant from French Guiana, which is genetically closer to and clusters with *M. liflandii* variant (named *M. liflandii*-like strain), could have evolved from the frog pathogen *M. liflandii* variant to infect humans (host-jump) [24]. This scenario needs to be studied further, notably by focusing on the activity and dating of IS*2404* and IS*2606* transposable genetic elements but remains an important step in our understanding of BU disease risk emergence. 

### 6.3. What about the Pathogenicity of Other Mycobacterium Species

Another important finding is the discovery of other mycobacterial species (not belonging to the MPM complex) associated with diagnosed BU cases. Stinear et al. [27] showed that out of 50 clinical samples diagnosed as BU, only 40 were positives for IS*2404*, raising the question about the etiology of the others. In Côte d’Ivoire, *M. chelonae* and *M. smegmatis* have been found in BU lesions [23] and more recently based on metabarcoding analysis of clinical samples, *M. gilvum* has been found associated with some BU cases in French Guiana [25]. For the latter cases, AFB (BAAR) staining was negative, IS*2404* and KR-B qPCR were both negative, and the culture on LJ solid medium was negative [25]. As we have shown that *M. gilvum* does not harbor IS*2404*, IS*2606* and KR-B in its genome and plasmids (Figure 1), this raises questions about its mechanisms of pathogenicity in humans. Taken together, these results raise the possibility that other mycobacterial species may have other types of virulence genes and immunosuppressive properties as the MPM variants that enable them to cause skin ulcerations similar to those caused by MPM variants. Furthermore, these results highlight the potential of metabarcoding studies, usually used for environmental investigations, in the search for the infectious agent responsible for human disease. These broader molecular screens could, for example, help the medical community target the most appropriate treatment and thus avoid late responses to inappropriate antibiotic therapy.

### 6.4. Implication for BU Treatment

The World Health Organization (WHO) recommends a combination of rifampicin (10 mg/kg once daily) and clarithromycin (7.5 mg/kg twice daily) to treat BU [75]. In Australia, although not proven effective, a combination of rifampicin (10 mg/kg once daily) and moxifloxacin or ciprofloxacin (400 mg once daily) is routinely used to treat patients (WHO, 2022). In Japan, a triple combination of rifampicin, levofloxacin and clarithromycin is used [76] while in French Guiana, treatment consists of a combination of rifampicin, amikacin or clarithromycin [77]. Whilst all based on rifampicin, BU treatment also combines other antimicrobial molecules that are variable upon the region. Rifampicin inhibits bacterial DNA synthesis through the inhibition of the DNA-dependent RNA polymerase [78]. However, some *M. ulcerans* variants previously showed resistance to rifampicin after monotherapy, suggesting that this mycobacterium has the ability to develop resistance against antimicrobial drugs commonly used to treat BU [79]. Whilst rifampicin and rifabutin have been shown to be the most active drugs against *M. marinum* variant infections [80] they are often treated with tritherapies including rifampicin, clarithromycin and ethambutol [81,82] thus slightly differing from *M. ulcerans* variant therapy. Amikacin, ciprofloxacin, kanamycin and lincomycin inhibited the growth of *M. pseudoshottsii* variant isolates [83]. In addition to being resistant to isoniazid, ethambutol and ethionamide, the *M. liflandii* variant also exhibits resistance to rifampicin (rifampin) and clarithromycin [84,85], suggesting that the combination of these later two antibiotics may not be appropriate to treat BU cases caused by *M. liflandii* variants. In French Guiana, *M. gilvum* infections were cured either after erysipelas treatment (based on penicillin administration) or with a combination of rifampicin (600 mg daily) and clarithromycin (500 mg twice daily) [25] while this mycobacterium has been shown to be resistant to isoniazid, sodium aminosalicylate and rifampicin [86]. These findings suggest that common combinations of antibiotic molecules locally used to treat BU might be inappropriately used depending on the variant/species causing disease. We suggest that future studies on BU should carefully identify the exact causative infectious agent in all diagnosed cases in order to target the most efficient antibiotic therapy.

## 7. Conclusions

Despite the extensive molecular evidence that all MPM represented variants rather than species and the association of other *M. ulcerans* variants and mycobacterium species with BU or BU-like cases, the *M. ulcerans* variant remained the sole targeted infectious agent. Numerous studies have used non-specific molecular markers to identify this MPM variant from patient samples and environmental matrices, with the aim of locating the source of human infection and tracking transmission dynamics. However, empirical evidence strongly suggests that the diversity of etiological agents responsible for BU has been underestimated and that other mycobacterial species that apparently do not produce mycolactone toxin have similar pathogenic characteristics in humans. The mechanisms underlying their pathogenicity remain to be elucidated but deserve to be examined by the medical and scientific community. Furthermore, while the phylogeny of these mycobacteria has been mainly genome-wide based, we suggest that another way of interpreting their evolution should focus on the copy number of ISs as well as on the pseudogenes present in their genomes, potentially suggesting evolutionary dynamics based on host adaptation rather than niche adaptation since *M. liflandii* and *M. ulcerans* variants have been found in the same (clinical and environmental) settings. Finally, we recommend that the diversity of *M. ulcerans* variants and/or mycobacterium species associated with BU or BU-like skin ulcerations should be carefully studied using broader molecular approaches such as metagenomics even for clinical samples and should be considered in the choice of the antibiotic therapy.

## 8. Material and Methods 

### 8.1. Literature Search

We systematically searched for all peer-reviewed journal articles studying the Buruli ulcer and/or *M. ulcerans* using the Web of Science and Google Scholar until 31 July 2020. We selected articles from the period 1996–2020 because the first PCR tests targeting *M. ulcerans* DNA were proposed from 1996 onwards. We excluded conference papers, preprints, PhD theses and papers that were not available online. We used the following keywords: (Buruli ulcer* OR Mycobacterium ulcerans*) AND (Mycobacterium ulcerans DNA OR detection) AND (PCR IS2404 OR PCR IS2606 OR PCR KR OR qPCR OR VNTR). We obtained a dataset containing 175 studies of which 89 were relevant in that they focused on *M. ulcerans* DNA detection or genetic characterization using molecular tests. We decided to exclude nested-PCR amplifications from our analysis due to their low use in the literature and their high potential to generate non-specific amplifications. Also, studies focusing on loop-mediated isothermal amplification (LAMP) were not included [87,88]. See “Studies” in Appendix A.

### 8.2. In Silico Evaluation of M. ulcerans Detection Primers

Thirty-one primer sets from nine previous reference studies were screened against the *M. ulcerans* genome and plasmid sequences to assess the efficiency of *M. ulcerans* detection methods. The Primer-Blast [89] and Primer3Plus [90] tools were used for this purpose. The complete genome sequence of *M. ulcerans* Agy99 (NC_008611.1) and the complete sequence of the Agy99 plasmid pMUM001 (NC_005916.1) were used as the database for the screening. Only primers with 100% alignment coverage and specific amplification (expected position and size) were considered effective. 

### 8.3. VNTR-Based Identification Key

Ten VNTR loci have been tested to construct the identification key. First, only 8 VNTRs (see “MIRU-VNTR” in the Appendix A) amplified by the primers validated in the in-silico evaluation step were retained. Then, two additional VNTRs (MIRU33 and VNTR18) for which the VNTR sequences are available and aligned against the Agy99 genome were added to the set. 

The VNTR sequences were extracted from the Agy99 genome based on the primers’ positions using an in-house extraction script. Next, the VNTRs sequences were aligned against an MPM genome variants database consisting of nine variants plus two outgroups, *M. marinum* and *M. gilvum* (Appendix A), using blastn implemented in BLAST+ suite [91]. To count the number of VNTRs in each genome, only matches with at least 99% alignment coverage were counted. 

The identification key was constructed with the VNTRs loci identified as the most discriminating for each variant. First, the loci were classified hierarchically according to the highest copy number variation (CNV) per locus. Then, the best combination of VNTR loci discriminating all variants is selected to finally classify the variants according to their CNV profile on the selected VNTRs.

### 8.4. Copy Number Variation of IS2404, IS2606 and KR

The target sequences for IS*2404*, IS*2606* and KR amplification were extracted from plasmid pMUM001 according to primer positions using an in-house extracting script. Next, the sequences were aligned against the MPM genomes and plasmids database consisting of nine variants plus two outgroups, *M. marinum* and *M. gilvum* (Appendix A), using blastn implemented in the BLAST+ suite [91]. To count the number of copy number variations in each genome and plasmid, only matches with at least 99% alignment coverage were counted. 

## Figures and Tables

**Figure 1 ijms-24-13727-f001:**
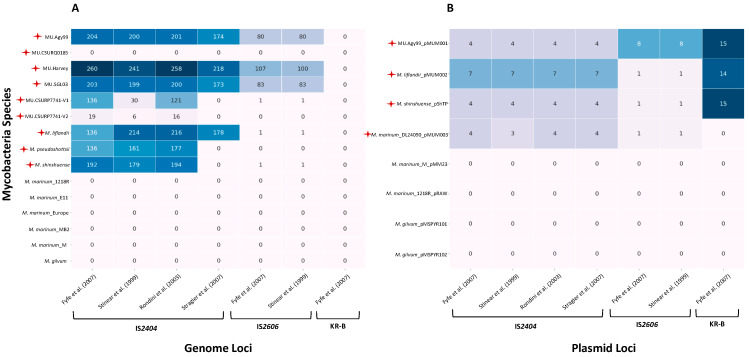
Presence and copy numbers of genetic elements IS*2404*, IS*2606* and KR-B in *M. ulcerans* (MU) strains and other MPM variants. *M. marinum* strains were selected as controls and *M. gilvum* was included as an outgroup but also as a mycobacterium not belonging to the MPM complex although it has recently been found in BU cases in French Guiana (Combe et al. 2020): [25]. (**A**) Mycobacteria selected according to the availability of their complete genomes. (**B**) Mycobacteria selected according to the availability of their plasmid. The most commonly used (q)PCR-based tests for *M. ulcerans* were selected from the list established in this study (See “Primers & Probes” in Appendix A). The presence of each molecular target in the genomes is indicated by the color gradient while copy numbers are shown for each strain/variant according to each molecular assay. Red stars indicate mycolactone-producing mycobacteria. Fyfe et al. (2007): [41]; Stinear et al. (1999): [27]; Rondini et al. (2003): [40]; Stragier et al. (2007): [54].

**Figure 2 ijms-24-13727-f002:**
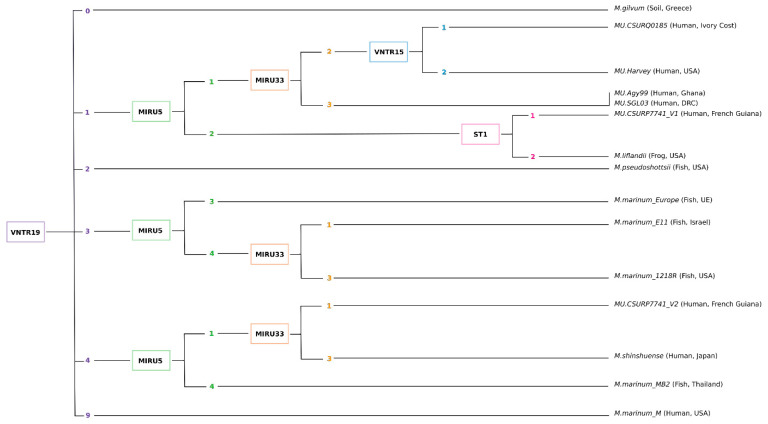
Universal key for identifying variants of MPM, *M. marinum* and *M. gilvum* based on a combination of previously published MIRU-VNTR loci. This key was established on the basis of the availability of complete genomes, for strains isolated from different geographical areas and allows the genetic distinction between MPM variants. *M. marinum* strains were included as controls (not belonging to the MPM complex) and *M. gilvum* was added as an outgroup (not belonging to the MPM complex) although found in BU cases in French Guiana [25]. The clustering process proposed here is based on the MIRU-VNTR loci DNA amplification and TRs copy number variations for VNTR19 (0; 1; 2; 3; 4 or 9), followed by MIRU5 (1; 2; 3; 4), followed by MIRU33, and so on. MU: *M. ulcerans*.

**Table 1 ijms-24-13727-t001:** Summary of the first isolation and phenotypic description of MPM. Adapted from the publication of Pidot et al. [19]. NA means not indicated.

MPM	First Isolation	Origin	Host	Colony Coloration	Growth Rate	Growth Temperature	Mycolactone	Source
*M. ulcerans*	1948	Australia	Human	Greenish or brownish yellow	Slow growth (>4 weeks incubation)	30 °C–33 °C	A/B	[13]
*M. shinshuense * (strain 753)	1980	Japan	Human	Yellowish, pigmentation in dark	Slow growth (>3 weeks incubation)	28 °C	A/B	[49]
*M. marinum* (strain CC240299)	2002	Israel	Koi (*Cyprinus carpio*)	NA	NA	NA	F	[48]
*M. marinum* (strain DL240490)	2002	Red Sea	European sea bass (*Dicentrarchus labrax*)	NA	NA	NA	F	[48]
*M. marinum* (strain DL045)	2002	Mediterranean Sea	European sea bass (*Dicentrarchus labrax*)	NA	NA	NA	F	[48]
*M. liflandii* (strain 128FXT)	2004	US	African tropical clawed frogs (*Xenopus tropicalis*)	Buff-colored, non-pigmented	Slow growth (>4 weeks incubation)	28 °C	E	[45]
*M. pseudoshottsii* (strain L15)	2005	US	Striped bass (*Morone saxatilis*)	Pale-yellow to gold, non-pigmented	Slow growth (>4 weeks incubation)	23 °C	F	[47]

## Data Availability

All data used are available as Appendix A and are published.

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
