# Peer review of "What about Current Diversity of Mycolactone-Producing Mycobacteria? Implication for the Diagnosis and Treatment of Buruli Ulcer"

_ijms, 2023, doi:10.3390/ijms241813727_

Round 1

Reviewer 1 Report

The review article by Combe et al. focuses on the classification and diagnosis of Buruli ulcer (BU) caused by M. ulcerans. Overall, the review discussed extensively topics including the origin of species assignment, molecular tools for diagnosis, and BU treatment. The method is well-documented. However, the authors failed to organize the vast amount of information and communicate it logically and succinctly. The manuscript, as a result, is hard to read. 

  1. The authors should emphasize the rationales for focusing on Mycolactone-producing mycobacteria species. 
  2. The writing and reasoning need to be more straightforward in 'section 2. On the origin of mycolactone-producing mycobacteria (MPM) species assignment', the author listed many discoveries chronologically but failed to organize and communicate the arguments logically. How they concluded that 'all MPMs should definitely be considered as variants from the same complex' remains unclear.
  3. The same problem exists in section 3; while the authors compiled many pieces of evidence, the information is not communicated clearly and succinctly. 

Author Response

#Referee1

Comment: The authors failed to organize the vast amount of information and communicate it logically and succinctly

Authors’ reply: We do not agree that our review is not organised (see details of the plan below), however after the reviewer’s comment we have reviewed and modified the order of a couple of section to strengthen the structure of the review. We believe it has now strengthen the clarity.

We accept that the review is dense with information, but excluding the introduction and conclusion, the information has been structured into 5 logical sections. The chronological logic we have followed to structure all this information is as follows: 1- A history of the diagnosis of this neglected tropical disease, i.e. The first diagnoses were purely clinical with all the possible margins of error for this type of ulcer, the appearance of the first diagnostic tests, first with PCRs and then with qPCRs; 2- The origin of the discovery of mycolactone-producing mycobacteria, which are the only ones responsible for pathology in humans. This section was presented before the history of diagnostic tools and following the reviewer's comments, we thought it would be more judicious to position it after the history of diagnostic tools, so we have moved it ; 3- Here we show that the new molecular diagnostic tools (section 1), which were used to determine the MPMs (section 2) are not specific, which is an important point given that the majority of BU cases reported to the WHO are based on these tests; 4- the awareness of epidemiologists of the need for more specific tools and the development of new tools, such as the MIRU-VNTR tools, to determine the diversity of M. ulcerans, and the realisation that these tools are no more specific than their predecessors; and finally, 5- the impact of this lack of diagnostic tools on the management of the disease, but also on its actual spatial and temporal distribution, and on the possibility of underestimating the number of BU caused by other pathogens with completely different environmental ecology.

Research on BU is carried out by a small number of groups of physicians and scientists worldwide and many have focused on identifying the causative agent and attempting to better characterise its ecological niche and mechanisms of pathogenicity. Our aim here was to synthesise the state of the art on the subject, pointing out that 1) all MPMs should be considered as variants of M. ulcerans rather than different species, which is not currently the consensus, and 2) the molecular tools most commonly used in published papers do not guarantee the unique presence of M. ulcerans in samples, nor that some skin lesions resembling BU could not be caused by other MPMs or even other mycobacterial species.

Our group has been working on BU for over 12 years, mainly on the identification of the biology and ecology of M. ulcerans, and we are part of the WHO BU expert group. So we know that such a comprehensive review is still lacking. As such, we believe that this review will be important to the medical and scientific community. Based on a comprehensive synthesis of the discovery of MPMs, it provides a rationale for species assignment and, more importantly, a fundamental blueprint for the medical and scientific community to better characterise the aetiology, ecology and biology of the disease.

THE PLAN

Introduction

  • History of BU diagnosis and ulcerans DNA testing

- Clinical and microbial diagnosis.

- First molecular diagnosis.

- Discovery and implementation of IS2404 and IS2606 PCR assays.

- Implementation of qPCR assays.

  • On the origin of Mycolactone-Producing Mycobacteria (MPM) species assignment
  • Lack of specificity of (q)PCR-based tests for ulcerans variant detection

- Lack of specificity of IS2404, IS2606 and KR-B.

- Lack of reliability of IS2404 and IS2606 CT-values.

  • Implementation of MIRU-VNTR to assess genotypic diversity

- Implementation of MIRU-VNTR typing.

- Main limitations and bias.

- The use of a universal identification key.

5- Implication for the diagnosis and treatment of BU

- Implication for BU diagnosis.

- Implication for M. ulcerans genetic diversity.

- What about the pathogenicity of other mycobacterium species?

- Implication for BU treatment.

Conclusions

Material & Methods 

- Literature search.

- In silico evaluation of M. ulcerans detection primers.

- Copy number variation of IS2404, IS2606 and KR.

Comment: The authors should emphasize the rationales for focusing on Mycolactone-producing mycobacteria species

Authors’ reply: This is an important point because it is the focus of our review, as only MPM species cause disease in humans. The rationale for focusing on MPMs is that acquisition of the mycolactone-producing plasmid was initially thought to be specific to M. ulcerans. However, a few years later, other mycolactone-producing mycobacteria carrying the virulence plasmid (mycolactone synthesis) were described. As there are currently major limitations and blinding errors with current molecular tools, this has direct and important implications for the treatment of this human disease, particularly as not all MPMs show exactly the same susceptibility/resistance to the variety of antibiotics available (see lines 807-833).

To clarify this in the manuscript, we have included some additional information in the Introduction section (lines 166-167, then from lines 176-178).

Comment: The writing and reasoning need to be more straightforward in 'section 2. On the origin of mycolactone-producing mycobacteria (MPM) species assignment', the author listed many discoveries chronologically but failed to organize and communicate the arguments logically. How they concluded that 'all MPMs should definitely be considered as variants from the same complex' remains unclear.

Authors’ reply: This section has now been moved - see answer above. For ease of reading, we have highlighted the rationale for this section from the outset (text in yellow), although the logic of our communication remains based on a chronological approach. We have summarised the historical discovery of MPM and the reasons for assigning different species names. We first introduced mycolactone as it was initially thought to be a characteristic of M. ulcerans. Then, in order to show that it was not a specificity of M. ulcerans, we had to present the discovery of other mycolactone-producing mycobacteria that had similar genomic and genetic characteristics. Here, based on previously published molecular evidence, we show that MPMs should not be considered as different species, but rather as variants. This has important implications for the epidemiology, diagnosis and treatment of the disease. We also draw an important conclusion regarding the appropriate terminology to use when referring to these MPMs.

Comment: The same problem exists in section 3; while the authors compiled many pieces of evidence, the information is not communicated clearly and succinctly. 

Authors’ reply: As suggested by the reviewer, we have now modified this section and added subtitles to make it easier to read, while maintaining the history of BU diagnosis. We believe that the chronology of BU diagnosis, from microbiological diagnosis to the introduction of PCR and qPCR assays targeting M. ulcerans, is the most logical. However, we have placed this section just after the introduction and before the section on the emergence of the MPM complex.

Reviewer 2 Report

The review is quite challenging to read due to its length and repetition of information. I recommend that the authors revise their manuscript for clarity. Additionally, it's important to adhere to general scientific conventions, such as italicizing species names, attaching figures directly to the manuscript, and providing proper figure captions.

You will find several comments in the attached file.

Author Response

#Referee2

Comment: The review is quite challenging to read due to its length and repetition of information. I recommend that the authors revise their manuscript for clarity. Additionally, it's important to adhere to general scientific conventions, such as italicizing species names, attaching figures directly to the manuscript, and providing proper figure captions.

Authors’ reply: As recommended by the reviewer, we have now restructured the manuscript, either by adding additional information to the text, or by adding subsections or reorganising paragraphs. We believe that these changes will make the reading clearer and more comprehensive. We agree that this is a long manuscript containing a lot of information, but since we are part of the WHO BU expert group working on the ecology and transmission of BU, we also know with certainty that such a descriptive and complete review of the history of MPM species assignment and the different molecular tools used to identify this pathogen is currently lacking. We are confident that such a complete literature review will be useful to the medical and scientific community, but also to young researchers starting to work on BU. Importantly, we also provide a framework for the biases and limitations in identifying this mycobacterium in biopsy and environmental samples. Such knowledge is essential if the medical and scientific community working on BU is to better understand its aetiology, ecology and biology.

We have gone through the manuscript carefully and italicised species names. We hope we have not missed any. When we submitted the manuscript, we had to include the figures as a .zip file, rather than including them directly in the text. However, as requested by the reviewer, we have now included the figures in the manuscript.

Could you please indicate why the figure captions are incorrect and how they should be changed?

All comments made in the .pdf file have been considered and included in the revised version of the manuscript.

Round 2

Reviewer 1 Report

The structure outlined by the authors was helpful in understanding the outline of each sub-topic. I am mostly satisfied with the revision. However, the first two paragraphs of the introduction seem distantly related to the review topic. And I suggest trimming them to better focus on BU and the causative bacteria.

Author Response

Author's reply:

We'd like to thank Rev1 for this feedback and we're pleased to see that the structure of the review is now more readable. As suggested, we have now reduced the first two paragraphs considerably.

Reviewer 2 Report

I want to thank the author for taking the time to meticulously reworking their manuscript and providing it in an improved format. I know this group is a leader in the BU field, so I was expecting a thorough review from them.

I see that the authors are concerned about my comment on the figure caption. I need to mention that in both versions they uploaded, I couldn’t find the figures they referred to (Fig. 1 and Fig. 2) for review. I only saw two supplementary figures. I did point this out when I first reviewed the manuscript, but the authors didn’t address it.

If the authors are dissatisfied with my review process, they have the option to request my withdrawal and approach the editorial board to assign a new reviewer for their manuscript.

Wishing you the best of luck.

Author Response

Author's reply:

We would like to thank Rev 2 for the very pertinent feedback on our review, which has helped to improve its readability.

On the contrary, we're very pleased with the comments and the analysis of our review. However, we don't quite understand why the figures are not accessible to the reviwer. We've asked the publishers to help us with this technical aspect. The figures were submitted separately and also included directly in the text. For our part, when we download the PDF version of the article, the figures are clearly visible. We have repeated this technical help to the editorial team.

Round 3

Reviewer 2 Report

Thank you for adding the figures to manuscript.